# Magnetic Contribution to the Seebeck Effect

**DOI:** 10.3390/e20120912

**Published:** 2018-11-30

**Authors:** Jean-Philippe Ansermet, Sylvain D. Brechet

**Affiliations:** 1Institute of Physics, Ecole Polytechnique Federale de Lausanne (EPFL), CH-1015 Lausanne, Switzerland; 2Physics Section, Ecole Polytechnique Federale de Lausanne (EPFL), CH-1015 Lausanne, Switzerland

**Keywords:** Seebeck, thermopower, thermoelectricity, magneto-thermopower, spintronics, 05.70.-a

## Abstract

The Seebeck effect is derived within the thermodynamics of irreversible processes when the generalized forces contain the magnetic term M∇B. This term appears in the formalism when the magnetic field is treated as a state variable. Two subsystems are considered, one representing atomic magnetic moments, and the other, mobile charges carrying a magnetic dipole moment. A magnetic contribution to the Seebeck coefficient is identified, proportional to the logarithmic derivative of the magnetization with respect to temperature. A brief review of experimental data on magneto-thermopower in magnetic metals illustrates this magnetic effect on thermally-driven charge transport.

## 1. Introduction

Thermoelectricity continues to be a subject of intense research. As energy is clearly a global issue, some scientists are motivated by the search for the largest possible Seebeck coefficient [1,2], and examine the potential of topological metals [3], while others seek to optimize the material ZT coefficient to attain a thermal efficiency which would be sufficient for widespread applications [4,5]. The anomalous Nernst effect is now an active field of research [6,7,8]. As a recent textbook indicates, there are still puzzles in thermoelectricity that need proper theoretical underpinning [9,10], though much progress has been made in describing transport in, for example, strongly correlated electron systems [11,12,13].

Thermoelectricity has also gained a lot of interest in fundamental research, thanks to the advent of spin caloritronics, the field of study of the cross effects between spin, charge, and heat transport. Intense research was sparked by the discovery of the spin Seebeck effect [14,15,16]. While the field has somewhat matured and reviews have appeared [17,18,19], its latest developments address, for example, time-resolved measurements [20], and new topics such as spin wave spin current [21], spin currents in antiferromagnets or in dichalcogenides [22,23].

Thermodynamics continues to be a method by which to investigate novel effects, including thermally-driven spin accumulation [24,25]. In this paper, we apply the thermodynamics of irreversible processes and find a connection between thermoelectricity and magnetization dynamics. The idea that magnetization dynamics contributes to thermopower is of course not new, but it was analyzed starting from a microscopic picture. Herzer, for example, analyzed the thermopower of amorphous ferromagnets [26]. When treating the case of temperatures near $T_{c}$, he included scattering of electrons with local magnetic moments, following the s-d model of Kasuya [27]. Thus, he found a contribution to the thermopower proportional to a correlation function of the local moments.

The use of correlation functions to calculate thermopower was described by Richter and Goedsche [28]. Their approach could account for experimental data on the intermetallics YCo${}_{12}$B${}_{6}$ and GdCo${}_{12}$B${}_{6}$ [29]. Korenblit used the s-d picture and Boltzmann transport equation to describe conduction in disordered ferromagnets. He inferred that the Seebeck coefficient reached a maximum at some temperature below $T_{c}$ [30]. Kettle et al. pointed out that the s-d model may be insufficient and that scattering by magnons should be taken into account [31]. These authors pointed out that the magnetic contributions could be better calculated in the framework of statistical thermodynamics of irreversible processes [32,33].

## 2. Thermodynamics with Magnetic Dipoles and Fields

When electromagnetic fields are treated as state variables [34], the entropy source density contains a term of the form
(1)$$\rho_{s}=\frac{1}{T}\sum_{\alpha }j_{\alpha }\cdot F_{\alpha }$$
where $j_{\alpha }$ designates either the vectorial entropy current density, or the current density of the substances *A*, *B*, … i.e., α=s,A,B,…. Here, we will consider two substances only, labelled *A* and *B*. Each substance carries a magnetic moment $m_{A}$ and $m_{B}$ per mole. If $n_{A}\left(r\right)$ is the number density of substance *A* at position r, then the magnetization of substance *A* is given by $M_{A}=n_{A}\left(r\right)m_{A}$ and likewise for *B*.

In order to ensure the positivity of $\rho_{s}$, we write
(2)$$j_{\alpha }=\sum_{\beta }L_{\alpha ,\beta }\;F_{\beta }$$
where the Onsager rank-2 tensors $L_{\alpha ,\beta }$ satisfy the condition
(3)$$\frac{1}{T}\left\{L_{\alpha ,\beta }\right\}\geq 0.$$

In Reference [34], we showed that in a metallic ferromagnet, in the absence of convection, intrinsic rotations and electric dipoles, $F_{B}$ is given by
(4)$$F_{B}=-\nabla {\bar{\mu }}_{B}+m_{B}\nabla B$$
where ${\bar{\mu }}_{B}=\mu_{B}+q_{B}V$ is the electrochemical potential of substance *B*, with $q_{B}$ the charge per mole of *B*, *V* the electrostatic potential, M the magnetization, and B the magnetic induction field. The notation F∇G is to be understood as, e.g., ${\left(F\nabla G\right)}_{x}=\sum_{i}F_{i}\partial_{x}G_{i}$. One consequence of this extra term in the generalized force is a damping term in the Landau–Lifshitz equation which is proportional to the temperature gradient [35]. The force term $m_{B}\nabla B$ is the equivalent for a continuum description to the force that a magnetic dipole experiences in a non-uniform magnetic induction field (Stern–Gerlach experiment). It is also known as the Kelvin force [36].

From Equation (Equation 2), we have the phenomenological transport equation
(5)$$j_{B}=-L_{Bs}\nabla T+L_{BB}F_{B}$$
where we have the usual contribution of the temperature gradient, $F_{s}=-\nabla T$, and a contribution of $F_{B}$ to the current $j_{B}$. We have omitted the cross-effect described by $L_{BA}$ for the following reason. Here below, we consider substance *A* as the core electrons which are responsible for the atomic magnetic moments of a ferromagnetic metal. Therefore, the magnetic moments of *A* are considered fixed and $L_{AB}$ can be assumed to vanish, since it describes the contribution of the generalized force of *B* to the current of *A*. Because of Onsager’s reciprocity relations, this implies also that $L_{BA}=0$.

## 3. Thermolectric Effect

Here, we want to analyze thermoelectric properties, namely, the Nernst and the Seebeck effects. The coefficients that characterize these effects are measured in the absence of current. Hence, we have $j_{B}=0$.

We assume that the magnetic induction field B varies spatially only because a temperature gradient is applied to the sample. We take the coordinate *z* along the temperature gradient. Thus we have
(6)$$\nabla T=\partial_{z}T\;\hat{z}$$
and
(7)$$m_{B}\nabla B=\left(m_{B}\cdot \frac{\partial B}{\partial T}\right)\partial_{z}T\;\hat{z}.$$

As is customary in thermoelectricity, we consider situations where the temperature changes over distances that are long compared to a characteristic relaxation length scale of substance *B*, such as a spin diffusion length. Then, there is no spatial variation of the chemical potential, $\nabla \mu_{B}=0$. Under these conditions, Equation (Equation 4) and Equation (Equation 5) yield
(8)$$\nabla V=-\frac{1}{q_{B}}L_{BB}^{-1}L_{Bs}\left(\partial_{z}T\;\hat{z}\right)+\frac{1}{q_{B}}\left(m_{B}\cdot \frac{\partial B}{\partial T}\right)\partial_{z}T\;\hat{z}.$$

Since the first term is expressed in terms of a rank-2 tensor, it may include a Nernst effet. On the other hand, the second term, which is due to the magnetic contribution to the force $F_{B}$, contributes only to the Seebeck effect.

Let us consider that substance *B* represents the conduction electrons of a ferromagnetic metal. The magnetic dipoles $m_{A}$ constitute the main contribution to the magnetization,
(9)$$M=n_{A}\;m_{A}.$$

By analogy with the s-d model [37,38,39,40], we assume from here on that the magnetic dipoles $m_{B}$ experience the exchange field due to the *A* electrons. Thus, we write
(10)$$B=\lambda_{x}\;M$$
where $\lambda_{x}$ is assumed to have the same value everywhere.

Since the spatial variation of B takes place over a large length scale, the magnetic moments $m_{B}$ track the magnetization M. When approaching the Curie point, the saturation magnetization $M_{s}=\left|M\right|$ depends on the temperature *T*. Therefore, we write
(11)$$m_{B}\nabla M=m_{B}\frac{\partial M_{s}}{\partial T}\partial_{z}T\;\hat{z}.$$

Under these conditions, Equation (Equation 8) implies a magnetic contribution to the Seebeck effect,
(12)$$\partial_{z}{V|}_{M}=-\frac{1}{q_{B}}\left(-\lambda_{x}\;m_{B}\frac{\partial M_{s}}{\partial T}\right)\partial_{z}T.$$

Therefore, this effect depends critically on how much the magnetization M depends on temperature. The corresponding Seebeck coefficient $\varepsilon_{M}$ is given by
(13)$$\varepsilon_{M}=\frac{\Delta E_{x}}{q_{B}}\left(\frac{-1}{M_{s}}\frac{\partial M_{s}}{\partial T}\right)$$
where $\Delta E_{x}$ is the exchange energy,
(14)$$\Delta E_{x}=m_{B}\lambda_{x}M_{s}.$$

We have used the usual sign convention for the Seebeck coefficient, according to which $\partial_{z}V=-\varepsilon_{M}\partial_{z}T$. As the exchange constant $\lambda_{x}$ in Equation (Equation 10) is positive, the magnetic contribution in Equation (Equation 13) to the thermopower is negative if $q_{B}<0$ (electrons), since $\partial M_{s}/\partial T<0$.

## 4. Experimental Evidence

Let us estimate the order of magnitude of the contribution in Equation (Equation 13) to the Seebeck coefficient. The exchange energy $\Delta E_{x}$ is of the order of 0.05 eV [38]. The logarithmic temperature derivative of $M_{s}$ can be estimated by assuming a temperature dependence of the form $M_{s}\left(T\right)=M_{s}\left(0\right)\sqrt{1-T/T_{C}}$ where $T_{C}$ is the Curie temperature. Thus, we write
(15)$$M_{s}\left(T\right)=M_{00}\sqrt{T_{C}-T}\;\Rightarrow \;-\frac{1}{M_{s}}\frac{\partial M_{s}}{\partial T}=\frac{1}{2}\frac{1}{T_{C}-T}.$$

At 90% of $T_{C}$, this implies $-\left(1/M_{s}\right)\partial M_{s}/\partial M_{s}=5/T_{C}$. If $T_{C}$ = 500 K, Equation (Equation 13) yields $\varepsilon_{M}$ = 0.5 mV/K.

A Seebeck coefficient was observed to peak near the Curie transition of a manganite and reached this order of magnitude [41].

A maximum in the thermopower at the Curie temperature was also observed in zinc-doped Nickel ferrites [42]. The maximum value was quite large, reaching (−5000 μV/K). However, the authors’ analysis brings out the possibility that this effect may be due to a peak in the charge density. Thus, this example serves as a warning that, while thermodynamics points to the possibility of a magnetic contribution to the Seebeck effect, one cannot take every maximum in the Seebeck coefficient at a phase transition as evidence of this mechanism.

A very large Seebeck coefficient(−5000 μV/K) was obtained for Nickel powders embedded in an insulating matrix, when the temperature was such that the thermal expansion made the material pass the percolation threshold of the material [2]. Unfortunately, the authors did not provide any data on magnetic susceptibility as a function of temperature, but it is likely to peak near the threshold [43].

So far, we have assumed a simple mean field model for the magnetization as a function of temperature. A first correction, which has been used for itinerant ferromagnets, consists in applying a correction to the exchange energy in Equation (Equation 14) of the form (Reference [44] (p. 245))
(16)$$m_{B}\lambda_{x}M_{s}=\Delta E_{x}\left(1-\alpha \frac{T}{T_{c}}\right).$$

The coefficient $\alpha^{-1}$ amounts to a corrective factor for the Curie temperature predicted by the Stoner model, given by $\alpha =D(E_{F})I-1$, where $D(E_{F})$ is the density of states at the Fermi level, *I* the exchange interaction of the Stoner model. Given the energy dependence in Equation (Equation 16), the maximum of the Seebeck coefficient is found at a temperature *T* given by
(17)$$\frac{T}{T_{c}}=\frac{1+\alpha }{2\alpha }.$$

Let us consider that the Stoner correction brings the predicted value for the critical temperature from $T_{C}$ to $T_{C}/2$, meaning that α=2. Then, we predict a maximum of the Seebeck coefficient at $T/T_{c}=3/4$. It turns out that the maximum is somewhat inferior to the estimate given by Equation (Equation 17). It was observed in a series of NiFeB alloys that $T/T_{c}$ ranged from 0.8 and 0.4 [31]. For Ni–Fe alloys, the maximum approaches a ratio $T/T_{c}$ closer to 1/3 (Reference [45], Figure 4).

## 5. Conclusions

The thermodynamics of irreversible processes is a powerful tool to discern a great variety of cross-effects in transport phenomena. It constitutes a framework within which to define a variety of spintronics effects [46]. Here, we make use of an earlier development in which we had treated the magnetic induction field as a state variable [34]. Thanks to this development, an additional term M∇B had been found in the generalized force associated with the matter current density.

Here, we show that this magnetic term implies a magnetic contribution to the Seebeck effect. When a temperature gradient is imposed on a sample, this magnetic term depends on ∇T provided the magnetization M depends on temperature. When we impose the condition of zero current corresponding to a thermoelectric power measurement, we find this term in the expression of the thermoelectric voltage.

We treat the case where the system is composed of two substances, *A* and *B*. In analogy with the sd-model for ferromagnetic metals, we assume *A* to be fixed, carrying a magnetic moment, and *B* to be mobile, experiencing the magnetic field due to *A*. We have assumed a vanishing contribution of the general force $F_{B}$ to the current of *A*, i.e., $L_{AB}=0$, since *A* represented fixed moments in our development.

We could extend our model further. If substance *A* was considered as supporting magnetization waves, it would make sense to include the cross term $L_{AB}$ in the calculation of the current $j_{B}$. This would allow for a magnon contribution to the Seebeck effect.

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
