# Peer review of "Magnetic Contribution to the Seebeck Effect"

_entropy, 2018, doi:10.3390/e20120912_

Round 1

Reviewer 1 Report

This paper deals with the contribution to the Seebeck effect possibly arising from the transport of magnetic moments in ferromagnetic materials. The method used in the the paper is the theory of non equilibrium thermodynamics where one makes the hypothesis that each current (of the extensive variables) is associated to a generalized thermodynamic force. For ferromagnets the identification of the generalized force associated to the magnetic moment current is not as obvious as for other physical quantities. The authors uses their own derivation of Ref.16, however other authors have made different choices. The topic is interesting, however the method used in the paper has several weak points that necessitates clarification.

1) The state of the art for what concerns the thermoelectricity on ferromagnets is not clear enough. Even less clear is the situation of thermodynamics of magnetic moment transport. The authors could improve the view on the topic.

2) The vector notation used for "M nabla B" (and probably for other similar terms) is not clear and creates confusion. The authors should use a notation commonly used in electromagnetism.

3) The physical meaning of "M nabla B" is also not clear. The energy of a magnetic moment m in a field is - mu0 m H, where H is the magnetic field generated by all sources apart from the moment itself. The authors use the field B instead. When extending to a ferromagnetic solid with a space dependent M(x), where B = mu0 (H+M), is it not clear what are the consequences. Please clarity this point.

4) The authors restrict themselves to an s-d model where there are different species of magnetic moments: A, the fixed d electrons, and B, the traveling s electrons. This choice is probably appropriate to model certain metallic ferromagnets, however is it some how confusing the whole picture because the reader is not able to distinguish between the results that are completely general from those that are specific to the AB (s-d) model. The authors can improve much the manuscript along this line.

Author Response

Attached is the reply to both reviewers.

Author Response

Attached are the replies to both reviewers.

Round 2

Reviewer 1 Report

I thank the authors for their responses and for the efforts done to improve their manuscript. Still, after the clarifications, I am not convinced that the force term of Eq.(4) is appropriate for ferromagnets. The problem comes, from my point of view, from Ref.[34] where in the expression (71) there is not the magnetic work term \mu_0 H dM. I am particularly concerned with Eq.(4) in connection with the work of Jonson and Silsbee (PRB, 1987) which is often used as a reference for thermodynamics of spin currents in ferromagnets and where the magnetic moment currents develops because of non equilibrium effects. Even if the thermodynamic approach developed by the authors can be of interest, I am not convinced that can be applied to the description of the Seebeck effect in metallic ferromagnets.